# Platelet-Released Factors: Their Role in Viral Disease and Applications for Extracellular Vesicle (EV) Therapy

**DOI:** 10.3390/ijms23042321

**Published:** 2022-02-19

**Authors:** Brita Ostermeier, Natalia Soriano-Sarabia, Sanjay B. Maggirwar

**Affiliations:** Department of Microbiology Immunology and Tropical Medicine, The George Washington University, 2300 I Street NW, Washington, DC 20037, USA; bostermeier@gwu.edu (B.O.); nataliasorsar@gwu.edu (N.S.-S.)

**Keywords:** platelets, microparticles, virus, vesicles, therapy, microvesicle

## Abstract

Platelets, which are small anuclear cell fragments, play important roles in thrombosis and hemostasis, but also actively release factors that can both suppress and induce viral infections. Platelet-released factors include sCD40L, microvesicles (MVs), and alpha granules that have the capacity to exert either pro-inflammatory or anti-inflammatory effects depending on the virus. These factors are prime targets for use in extracellular vesicle (EV)-based therapy due to their ability to reduce viral infections and exert anti-inflammatory effects. While there are some studies regarding platelet microvesicle-based (PMV-based) therapy, there is still much to learn about PMVs before such therapy can be used. This review provides the background necessary to understand the roles of platelet-released factors, how these factors might be useful in PMV-based therapy, and a critical discussion of current knowledge of platelets and their role in viral diseases.

## 1. Background

Platelets are anucleate cytoplasmic cell fragments 2–4 μm in diameter that play important roles in regulating blood hemostasis and thrombosis [1]. In the event of vascular injury, platelets, along with leukocytes and red blood cells, activate a coagulation cascade to arrest blood loss and restore hemostasis [2,3]. Platelets originate from proplatelets, which are evaginations of megakaryocytes, a bone marrow resident cell. Megakaryocytes are derived from a common myeloid progenitor cell but are distinct from lymphocytes. Platelets migrate to the plasma, where they are predominately found [4]. As platelets are merely fragments from megakaryocytes, this explains why they are anucleate. However, despite being anucleate, platelets contain messenger RNA (mRNA) and are able to produce proteins and they can endocytose proteins from their environment. Even without a nucleus, platelets can actively interact with cells and their environment, directly influencing cellular functions.

In addition to platelets’ regulation of hemostasis, platelets also play an important role in modulating both innate and adaptive immune responses. Platelets express innate immune cell pattern recognition receptors (PRRs), such as toll-like receptors (TLRs) 1, 2, 4, 6, 7, 8, and 9 [5,6,7,8]. Therefore, like other PRR-expressing cells, platelets can recognize pathogen associated molecular patterns (PAMPs) and perform functions, such as the expression of inflammatory cytokines, upon pathogen recognition. Platelets also express adaptive immune cell receptors, such as major histocompatibility complex (MHC)-I and CD40L. Thus, platelets can induce CD8+ T cell activation through the expression of MHC-I, mediate peripheral blood B cell activation, and enhance dendritic cell responses through CD40L–CD40 interactions [9,10,11]. Therefore, platelets are a unique and diverse set of cell fragments that perform a wide range of functions. However, for platelets to perform these diverse immune effector functions, they must first become activated.

Viruses are characterized as one of the activators of platelets. Upon activation, platelets release a multitude of factors and extracellular vesicles. These extracellular vesicles deliver platelet particles to surrounding cells and perform anti-viral and pro-inflammatory functions among others. While many of the functions are still not completely understood, current extracellular vesicle therapies leverage the anti-viral functions of extracellular vesicles to treat viral diseases. However, the pro-inflammatory effects of platelet-derived extracellular vesicles are at times detrimental to viral clearance, resulting in the progression of the viral infection to severe disease states. Due to the duality of these two functions, it is important to understand the molecular mechanisms surrounding platelet-released factors and viral diseases to be able to better utilize platelet extracellular vesicles as a form of therapy.

In this paper we review current knowledge on platelet immunobiology and effector functions and current literature on platelet interactions with specific viral diseases, such as human immunodeficiency virus-1 (HIV-1), dengue virus (DENV), influenza A virus (IAV), and severe acute respiratory syndrome coronavirus 2 (SARS-CoV-2). Additionally, in this paper we review current techniques for extracellular vesicle therapy and speculate as to how platelet microvesicles might be employed as a therapeutic tool in viral diseases.

### 1.1. Platelet Activation

Platelets exist in two main “states”: a resting state and an activated state. Most platelet effector functions occur after platelets are activated. There are three main pathways for platelet activation: immunoreceptor tyrosine-based activation motif (ITAM) signaling, G protein-coupled receptor (GPCR) signaling, and toll-like TLR-mediated signaling. ITAM signaling is generally associated with immune effector functions, GPCR signaling is associated with thrombosis, and TLR signaling is associated with both hemostatic responses and immune responses [12]. Glycoprotein (GP) Ib-IX-V, GP VI, and C-type lectin-like receptor 2 (CLEC-2) all act through ITAM signaling, while soluble agonists released by activated cells, such as ADP, thromboxane A_2_ (TxA_2_), and thrombin, act through GPCR signaling [13,14,15,16]. TLR4 signaling occurs via the binding of lipopolysaccharide (LPS), histones, HMGB1, Fn-EDA+, and viral proteins, such as dengue virus NS1 (Figure 1) [12,17]. Intracellular TLR7 signaling is induced from viral ssRNA (Figure 1) and increases platelet neutrophil interactions [18]. TLR3 is also expressed by platelets and plays a role in platelet activation in the presence of viral dsDNA (Figure 1). All three pathways for platelet activation lead to degranulation and the expression of integrin adhesion receptors, such as CD62P (P-selectin) [12,14,19,20]. Delivery of immunomodulatory molecules within granules and increasing aggregation/complexing with other cells, initiates the platelet effector functions.

Upon platelet activation, CD62P, which is stored in alpha granules, is transferred to the plasma membrane of the cell. CD62P is a widely recognized marker of platelet activation and is known to bind P-selectin glycoprotein ligand 1 (PSGL-1) on the surface of other cells, which forms what are known as platelet-cell complexes. Many of these platelet-cell complexes include platelet-leukocyte complexes that help mediate leukocyte rolling to initiate extravasation into other tissues [21]. Extravasation is thought to occur through platelet P-Selectin binding to PSGL-1 on leukocytes, which leads to ERK1/2 MAPK expression and changes to leukocyte integrin expression, which are known cell mediators of extravasation [21]. Platelets may mediate leukocyte homing to lymph nodes [22]. In vitro, activated platelets, bind to circulating lymphocytes and to the peripheral node addressin (PNAd), and mediate rolling in high endothelial venules all via CD62P, which is expressed on the platelet’s surface [22].

#### Virus Entry and Activation in Platelets

Viral binding to platelets can trigger platelet activation, as is seen in human immunodeficiency virus 1 (HIV-1) and dengue virus (DENV) (Figure 1). SARS-CoV-2 binds to platelet surface ligand angiotensin-converting enzyme 2 (ACE2) as shown in Figure 1, but it is not yet known what type of platelet activation response is induced by the viral binding of ACE2 [23]. However, for some viruses, such as influenza, platelet internalization of the virus triggers activation rather than viral binding. For influenza A virus (IAV), internalization of viral particles into the open canicular system (OCS) within platelets, leads to endosomal toll-like receptor (TLR)-mediated platelet activation (Figure 1).

HIV-1 can bind to platelets through multiple receptors, including chemokine receptor CXCR1, 2, and 4, CCR1, 3, and 4, as well as via C-type lectin-like receptor 2 (CLEC-2) and dendritic cell specific intercellular adhesion molecule (ICAM)-grabbing non-integrin (DC-SIGN) (Figure 1) [24]. This binding leads to platelet activation, which is strongly correlated with HIV-1 viral load [24,25]. One HIV-1 protein, trans-activator of transcription (Tat), is specifically responsible for platelet activation through β3 integrin and CCR3, which leads to the secretion of CD40L and the expression of CD62P [26]. Because CD62P is involved in platelet-leukocyte complexes, an increase in the secretion of CD62P also increases platelet complexing. We and others have shown that CD62P expressing platelets can enhance the transfer of infectious HIV from platelets to CD4+ T cells through platelet-T cell complexes [27].

DENV engages with DC-SIGN and heparan sulfate proteoglycans, which mediate platelet internalization of DENV and activate platelets (Figure 1) [28]. A second method of platelet activation via DENV is through DENV binding to CLEC2, which leads to platelet activation and the secretion of EVs (Figure 1) [29]. A third method of DENV activation of platelets is DENV nonstructural protein 1 (NS1) binding to TLR4, which increases platelet aggregation, adherence to endothelial cells, and phagocytosis by macrophages (Figure 1) [30]. NS1 is required for inflammatory signaling in dengue infected platelets [31]. DENV-infected platelets also correlate with an increase in CD62P surface expression [32]. Activation in the presence of DENV is also characterized by thrombocytopenia, or reduced platelet counts. Activation of platelets by DENV is also coupled with the expression of apoptosis markers, such as milk fat globule-epidermal growth factor 8, TIMD4, and CD36, which leads to platelet phagocytosis by monocytic cells and thus clearance from the plasma [17,33]. The degree to which DENV induces platelet activation can determine the severity of thrombocytopenia [17]. Since thrombocytopenia is a marker of DENV disease severity, it is reasonable to conclude that platelet activation may be an important determining factor in DENV disease severity.

IAV single stranded RNA (ssRNA) binds to TLR7 within the endosome of platelets, triggering platelet activation (Figure 1) [34]. Internalization of IAV appears to be rapid (<30 min) [34]. This internalization likely activates platelets early on in infection. In accordance with the ability of IAV to activate platelets, influenza patients showed increased levels of activated platelets compared to age- and sex-matched healthy controls [35].

The connections between virus binding, activation of platelets, and (in the case of DENV) clearance of platelets are still under investigation. Whereas viruses can activate platelets, platelet activation also occurs in absence of viral particles due to surrounding pro-inflammatory cytokines from nearby cells. Thus, the source of platelet activation might also play a role in platelet effector functions and disease progression. Platelets are activated through multiple different ligands; thus, the pathways for platelet activation may play a role in host clearance of pathogens. Finally, one of the remaining questions surrounding platelet activation is: how much activation is too much? The fine line between a “healthy” level of platelet activation and a “damaging” level of platelet activation is a balance that remains to be fully understood.

## 2. Platelet Effector Functions

Once a platelet is activated, it exerts many effector functions, but we focus on the secretion of CD40L and release of extracellular vesicles (EVs) and granules (Figure 1). These effector functions allow for increased adhesion of platelets to monocytes and leukocytes, increased monocyte and leukocyte extravasation, and monocyte and leukocyte migration. In some cases, increased clearance of viruses occurs; in other cases, reduced clearance of viruses is noted. Understanding the major molecules and structures released by activated platelets can provide insight into the role that platelets play in viral disease progression and inspire new therapeutic approaches.

### 2.1. Expression and Release of CD40L

CD40L (CD154) is a ligand expressed by T cells, which interacts with CD40 on B cells and is essential to humoral development and isotype switching [36]. However, CD40L is also expressed by platelets whereas CD40 is expressed on monocytes, macrophages, dendritic cells, and endothelial cells. CD40L expressed on platelets can interact with CD40 expressed on endothelial cells, leading to a local release of adhesion molecules ICAM1, vascular cell adhesion molecule 1 (VCAM1), and CCL2, triggering inflammatory immune reactions [37,38,39].

Upon platelet activation, in addition to the surface expression of CD40L, soluble CD40L (sCD40L) is released. Platelet-released sCD40L accounts for most of the sCD40L in circulation and can act both on nearby cells (paracrine) and on the platelets that secrete them (autocrine). Similar to membrane bound CD40L, sCD40L can interact with CD40 expressed on vascular cells, such as endothelial cells [40]. sCD40L–CD40 interactions on endothelial cells can enhance the expression of adhesion molecules, such as P-selectin and E-selectin, and lead to the release of tissue factor and IL-6 [41,42]. CD40L induced adhesion molecule expression promotes complex formations, such as that which occurs between platelets and monocytes [43].

#### Role of CD40L and sCD40L in Viral Infections

CD40L plays a role in viral infections both as plasma membrane bound CD40L and as secreted sCD40L. However, most studies report on sCD40L, likely due to the ease of its detection in the plasma. HIV-1, DENV, IAV, and SARS-CoV-2 infections have each been linked to sCD40L levels in the plasma (Figure 2). This has important implications for the role that platelets play in disease progression.

HIV-1 Tat mediated activation of platelets leads to the secretion of sCD40L [26]. In the presence of CD40L, HIV-1-Tat -injected mice exhibit increased leukocyte adhesion to the brain microvasculature [44]. This suggests that Tat-induced secretion of CD40L mediates leukocyte adhesion at the blood brain barrier, which may mediate leukocyte trafficking into the brain (Figure 2). Additionally, sCD40L is elevated in the plasma of people living with HIV-1 (PLWH) and may induce immunosuppression [45,46]. This immunosuppression is characterized by increased regulatory T-cell differentiation and is likely an evolutionary mechanism to reduce T-cell-effector killing of HIV-infected cells (Figure 2). The current therapies for HIV-1 reduce sCD40L levels in the plasma. For PLWH that are naïve to combined antiretroviral therapy (cART), that plasma sCD40L level was significantly elevated compared to that in cART-treated PLWH and healthy controls [47]. This suggests that HIV-1-induced sCD40L release is attenuated by the administration of cART (Figure 2).

sCD40L levels in dengue patient plasma are also elevated. sCD40L increased by almost five-fold 2 h post interaction and two-fold 4 h post interaction with DENV-2 [48]. This indicates that DENV can trigger sCD40L release in platelets. However, differing from HIV, this sCD40L release may not be maintained in cases of severe disease, which are characterized by thrombocytopenia [49]. Patients with severe dengue indicated by plasma leakage had significantly reduced sCD40L compared to the dengue cases without plasma leakage and uninfected control individuals (Figure 2) [50]. In cases of non-fatal viral infection, such as Ebola hemorrhagic fever, sCD40L levels are increased compared to fatal cases [51]. A positive correlation exists between sCD40L levels and platelet counts, indicating that platelets are a primary and necessary source of sCD40L levels in the plasma during viral infection [50]. Thus, the maintenance of platelet counts leads to continued secretion of sCD40L, which may be an important factor in viral disease severity and fatality (Figure 2).

SARS-CoV-2 infection is also characterized by platelet activation, expression of CD40L, and the presence of sCD40L in the plasma [52,53]. However, similar to dengue, severe infection can be characterized by thrombocytopenia and one study found reduced sCD40L levels [52,54,55]. However another study found that ICU vs. non-ICU COVID-19 cases have higher sCD40L levels [53]. More research regarding sCD40L levels and disease severity are needed to determine the full effect of sCD40L on disease progression for those with severe COVID-19. sCD40L levels are found to be associated with severe COVID-19 manifestations, such as myocardial dysfunctions, mortality, and vascular biomarkers in SARS-CoV-2 patients [53,56]. While sCD40L levels and platelet activation have been associated with the COVID-19 disease, hospitalizations, and other SARS-CoV-2 complications, it is still uncertain what role CD40L plays in disease progression and severity (Figure 2) [57].

Similar to DENV and SARS-CoV-2, severe IAV has also been associated with thrombocytopenia [58,59]. Plasma sCD40L has been linked to the immunopathology of IAV infection [60,61]. Hospitalized influenza patients’ plasma sCD40L levels were correlated with influenza associated encephalopathy (Figure 2) [62]. One study found that extracellular histones bind to platelets and trigger the release of sCD40L in influenza virus infection [60].

In the above viral infections, sCD40L is an important serum marker of infection and, in some cases, for the poor prognosis of disease progression. However, there appears to be differences in whether sCD40L exists in the serum during severe disease, and this is likely due to the level of thrombocytopenia. Future research will be needed to investigate this connection and elucidate the mechanisms involved in inflammatory lung diseases, such as SARS-CoV-2 and influenza.

### 2.2. Platelet Extracellular Vesicle Release

Similar to immune cells and endothelial cells, platelets release extracellular vesicles (EVs). Although EVs are implicated in disease progression, they are also promising therapeutics, including as vessels for drug delivery. There is variation in the usage of the notation “EVs”. EVs can be used to describe membrane vesicles (1–5000 nm) with phospholipid bilayers that are secreted by a cell. Multiple cell types secrete EVs that fit this size specification, including platelets. The International Society for Extracellular Vesicles recommends to use the term “extracellular vesicle” (EVs) as an umbrella term from which vesicles can be further characterized [63]. These subdivisions of EVs include: exosomes (30–100 nm), microvesicles (MVs) (100–1000 nm), and apoptotic bodies (1–5 µm) [64]. MVs secreted by platelets are generally referred to as “microparticles” or “MPs”. However, this notation is not adopted in all papers. Some papers use the term “microvesicles” or “MVs” to refer to both exosomes and shedding vesicles [65]. For the purpose of this paper, we use the term “microvesicle” (MV) to refer to platelet-released particles 100–1000 nm in diameter.

MVs are referred to in the literature through multiple abbreviations, such as platelet extracellular vesicles (pEVs) and platelet microparticles (PMPs). MVs range from 100 to 1000 nm in diameter, but tend to be around 200 nm (larger than exosomes which are between 30–100 nm in diameter) [66]. In the blood, MVs derived from megakaryocytes and platelets are the most abundant, making up an estimated 70–90% of the microvesicles in the bloodstream, including in healthy individuals [67,68,69]. It is still unclear what the source of MVs is in healthy individuals, but it is suggested that MVs in healthy individuals originate from megakaryocytes, which are constitutively expressing MVs [70]. Megakaryocyte derived microvesicles (MKMVs) are CD41+ and express phosphatidylserine (PS) on their surface similar to platelet-derived microvesicles (PMVs). PMVs and MKMVs can expose PS on their outer membrane upon activation, although this process has yet to be elucidated [71]. Some papers only report PS+ MVs, but it is important to note that not all MVs express PS on their surface; thus, MVs are a heterogenous population [72]. Whilst MKMVs are also interesting MVs that are worth of study, for the purpose of this review we focus on PMVs.

PMVs are a subset of platelet EVs that are released from platelets through blebbing. They retain a plasma membrane often surrounding platelet-derived RNA, such as microRNA (miRNA) or messenger RNA (mRNA), mitochondrial components, and proteins. The presence of a plasma membrane and RNA separates them from granular microparticles, where the granular contents are released upon fusion with the OCS of platelets. This is a specific distinction made in this review article and we hope that future research in the field will follow suit in order to more consistently and correctly categorize PMVs.

PMV-derived miRNAs can be incorporated into target cells and induce signaling [73]. Importantly, there are lipoprotein-like nonvesicular particles that contain RNAs; thus, PMVs are not the only source of extracellular RNAs circulating in the blood [74,75]. Future research should make a distinction between PMV-derived RNAs and non-PMV-derived RNAs. Previous efforts to make a distinction between PMVs and MKMVs might be useful in making the distinction between liposomal particles and PMVs [70].

Other than the presence of RNA, PMVs carry other, very similar factors, to alpha granules, such as cytokines/chemokines, growth factors, and transcription factors [76,77]. PMVs can also contain damage associated molecular patterns (DAMPs), such as high-mobility group protein 1 (HMGB1), S100A8/9, mitochondrial DAMPs, and oxidation-specific epitopes [78,79,80,81,82].

#### The Role of PMVs in Viral Infections

Many viruses are involved in the initiation of platelet activation and since activation of platelets leads to MV release, it is plausible that, in a viral infection, viruses themselves can trigger the release of PMVs. The presence of PMVs in patient serum is an indicator of severe disease in respect of many of the viral diseases mentioned below. PMVs are also implicated in disease progression through the specific cargo and receptors that they carry.

HIV-1 Tat binds to the chemokine receptor CCR3 and b integrin on platelets, leading to the release of PMVs [26,83]. Platelet derived extracellular vesicles can be used as inflammation markers and immune activation markers in HIV infection and are related to procoagulant activity [84,85,86,87]. Characterization of circulating microvesicles through flow cytometry indicates that most of circulating MVs are PMVs (79%) in HIV infection and that these MVs have significantly lowered mitochondrial densities compared to uninfected controls [83]. The implications of mitochondrial density in PMVs is unknown; however, long-term treated HIV is associated with reduced platelet mitochondrial content, supporting the presence of MVs with lowered mitochondrial densities [88]. Circulating MVs, most of which are platelet-derived, were also implicated in adversely affecting endothelial cells in an in vitro model [87]. Among many HIV-1 receptors, PMVs express CXCR4. CXCR4 expressing platelets can transfer CXCR4 to CXCR4 negative cells through membrane fusion [89], indicating a role of PMVs in X4 HIV-pathogenesis (Figure 3).

DENV activates platelets via CLEC2, leading to PMV release, which typically exist at elevated levels in patient plasma with procoagulant activities [29,90]. These PMVs are shown to enhance macrophage and neutrophil effector functions, including neutrophil extracellular trap (NET) formation and pro-inflammatory cytokine release (Figure 3) [29]. DENV-induced NET formation and pro-inflammatory cytokine release reduced mouse survival in vivo (Figure 3) [29]. Additionally, DENV patients with thrombocytopenia without bleeding had significantly more PMVs than DENV patients with thrombocytopenia with bleeding [91]. These findings indicate that DENV-mediated release of PMVs may play an important role in disease progression and might be a novel biomarker for DENV patients (Figure 3).

IAV also induces PMV release due to platelet activation [92]. PMV release is stimulated in platelets by the formation of thrombin and by the binding of H1N1 to FcγRIIA [92]. Similar to some reports of DENV, influenza-related mortality correlates with elevated plasma levels of TF-expressing PMVs (Figure 3) [93]. Additionally, thrombocytopenia is associated with increases in influenza virus pathogenicity and is dependent upon the subtype of influenza virus [94]. Unlike DENV and HIV-1, there is still much unknown about the PMVs involved in IAV infection and disease progression.

Similar to IAV, SARS-CoV-2-associated mortality is correlated with elevated plasma levels of TF-expressing PMVs; however, PMVs are not alone in TF expression as peripheral blood mononuclear cells (PMBCs) also express elevated levels of TF [93,95]. Elevated plasma levels of TF-expressing PMVs is also associated with COVID-19 disease severity and thrombosis [93]. Similar to HIV, IAV, and some DENV cases, SARS-CoV-2 has been found to be associated with increased levels of circulating PMVs and EVs (Figure 3) [96,97,98,99,100]. Interestingly, one study found that severe COVID-19 cases and healthy individuals had reduced phosphatidylserine (PS) exposing PMVs compared to moderately infected individuals [96]. However, another study found that increasing PS, PMV, and peripheral blood mononuclear cell (PBMC) frequencies were found to be strongly correlated with increasing COVID-19 disease severity [101]. In either case, it appears that PS+ PMVs are biomarkers of at least moderate SARS-CoV-2 infection. In the debate on whether severe COVID-19 cases exhibit increased PS+ PMVs, it might be important to determine what constitutes severe COVID-19 and whether it encompasses severe thrombocytopenia.

PMVs play diverse roles in viral disease and many of their roles have yet to be elucidated. PMVs might lead to increased disease severity in some viral diseases, but attenuate other viral diseases. Understanding more about PMVs is crucial to understanding more about these viral infections and disease progression.

### 2.3. Degranulation

Platelet degranulation occurs upon platelet activation. Granules include dense granules, lysosomes, and α-granules, which contain various factors involved in coagulation, inflammation, and antimicrobial host defense. These factors can be pre-synthesized from megakaryocytes, platelet-synthesized factors, and endocytosed factors trafficked to the granules. Activated platelets can release granules by causing the fusion of platelet granules with the OCS, leading to the release of granule contents into the extracellular environment [102,103]. Dense granules released by platelets contain bioactive amines (serotonin, histamine), phosphates (polyphosphate, pyrophosphate), cations, ADP, and ATP, which are generally involved in hemostasis and thrombosis and potentiate platelet activation [104,105]. Lysosomes released by platelets contain degrading enzymes (collagenase, elastase, cathepsins), carbohydrate degrading enzymes (glucosidase, galactosidase), and phosphatases, which are all generally involved in degradation [104].

α-granules (200–500 nm in diameter) are the most abundant granule in platelets with approximately 50–80 α-granules per platelet [106]. α-granules have membrane bound proteins that are also on the plasma membrane of platelets, such as integrins, immunoglobulin family receptors, leucine-rich repeat family receptors, tetraspanins, and other receptors [105]. α-granules also contain transporter factors associated with antigen processing (TAP) molecules and an entire proteasome system, preserving their ability to process and present peptide on major histocompatibility complex I (MHC I) [107,108,109].

α-granules have three main functional roles: coagulation, inflammation, and antimicrobial host defense [105]. Factors involved in coagulation include primary hemostatic factors, such as von Willebrand factor (vWf), GPIbα-IX-V, integrins, major fibrinogen receptor, collagen receptor, and GPVI, which contribute to platelet adhesion [110,111,112]. Secondary hemostatic factors in α-granules include Factors V, XI, and XIII [113]. Hemostatic balance is also maintained through α-granule factors antithrombin, C1-inhibitor, factor XIIa, TFPI, protein S, and protease nexin-2 [105,114,115,116]. α-granules also contain chemokines that are primarily involved in recruiting innate immune cells, leading to increased inflammation. However, α-granule chemokines can also stimulate/suppress the proliferation of immune cells and alter immune cell phenotypes. Platelets are a major cellular source of CXCL4 and CXCL7, which are the most abundant chemokines in α-granules and, subsequently, the most studied [117]. CXCL4 is known to be a chemoattractant for monocytes and neutrophils; it can limit neutrophil and monocyte apoptosis, which promotes their survival, stimulates the proliferation of regulatory T cells, but inhibits the proliferation of other T cells, and it can alter monocyte phenotypes [118,119,120,121,122]. CXCL7 is proteolytically cleaved into platelet basic protein (PBP), β-thromboglobulin, and CTAPIII, which bind receptors CXCR1 and 2, prompting neutrophil and endothelial progenitor cell migration [123]. Finally, α-granules contain antimicrobial proteins, such as the previously mentioned chemokines CXCL4, CXCL7, and CCL5 [124,125], indicating both the anti-inflammatory and the anti-microbial properties of these cytokines.

α-granules contain diverse cargo including cytokines/chemokines, growth factors, angiogenic factors, anti-angiogenic factors, and hemostatic factors and some of these factors carry out multiple different functions. The above listed factors are known to be present in α-granules, however there are many platelet-released factors that are released from platelets but are not yet confirmed to be released from α-granules [104,105,112].

#### The Role of α-Granules in Viral Infection

It is plausible that when HIV-1 or other viruses are within the OCS of platelets, α-granules can fuse with the OCS to deliver cargo aimed at inhibiting the virus [126]. There are numerous factors within platelet α-granules; however, CXCL4 appears to be the most linked to viral infection.

CXCL4 or platelet factor 4 (PF-4) is a chemokine derived from platelet α-granules, but it is also present in PMVs (Figure 4). Its role in viral infection appears to be beneficial in protecting against most viruses. α-granule derived CXCL4 (PF4) directly inhibits HIV-1 by binding the envelope protein, gp120 (Figure 4) [127]. However, it was later found that the protective effect of CXCL4 is present when CXCL4 is in a monomeric form [128]. Upon increasing CXCL4 concentration, oligomerization occurs and promotes viral replication in vitro [128]. Another study confirmed findings that CXCL4 is protective against HIV-1 infection, specifically in T cells (Figure 4) [129]. Decreased CXCL4 serum levels are correlated with poor SARS prognosis, indicating the importance of CXCL4 in protection from severe SARS (Figure 4) [130]. Similar to HIV-1 and SARS, CXCL4 was found to protect mice from H1N1 infection [131]. However, this protection against H1N1 is specifically linked to the recruitment of neutrophils to inflamed lungs (Figure 4) [131]. CXCL4 is an abundant protein in dengue patient plasma and it is a prognostic tool for predicting acute vs. severe dengue (Figure 4) [132,133]. CXCL4 is indicated as a significant factor in DENV replication because treatment with anti-CXCL4 decreases pro-inflammatory cytokines TNF-α, IL-1β, and IL-6 in monocytes and rescues the cells from infection in vitro [132].

Therefore, among all four viral diseases discussed, CXCL4 plays a major role in viral infection. In some cases, such as HIV, SARS, and H1N1, CXCL4 within α-granules can be protective against infection. However, in both HIV and DENV, CXCL4 within α-granules can promote infection. Thus, CXCL4 is a multifaceted factor that should be investigated further in the context of each virus to determine potential therapies based on the mechanism of action of CXL4 within α-granules.

## 3. Therapeutic Applications

Platelet-released factors during viral infection hold therapeutic potential due to their relatively small size and potent effects on disease progression. A desirable vehicle for delivering these factors is platelet-released extracellular vesicles (pEVs), which include PMVs, platelet exosomes (Exos), and platelet apoptotic bodies. pEVs could be engineered to express platelet-specific receptors, to contain specific platelet factors, or to contain RNA, which could all potentially be delivered to target cells (Figure 5). Given what we currently know about the role of platelet-released factors in viral disease, it is feasible that the development of pEV-based therapy for viral diseases lies in the near future.

### 3.1. Advantages of pEV Therapy

One of the most promising aspects of pEV therapy is the ability of EVs to cross biological barriers, such as the blood–brain barrier and synovial membranes [134,135]. Platelet derived pEVs have not yet been confirmed to cross the blood–brain barrier. However, pEVs have been found in the synovial fluid associated with inflammation and rheumatoid arthritis (RA) [136]. pEVs can also reach lymphoid organs, such as the spleen and lymph nodes, with some ability to enter the bone marrow, liver, and lungs, as found in an ovalbumin-based mouse model [136]. During inflammation, pEVs traffic to the bone marrow, influencing differentiation of the megakaryocytes that reside there [137]. However, it is important to keep in mind that EVs found within the brain, synovial fluid, and bone marrow are all associated with inflammation. In severe virally infected individuals with uncontrolled inflammation, administration of pEVs should be carefully considered because it may be hard to control just where and how these vesicles act. Much more research is needed for these sites.

Differing from the other biologically restricted areas, pEVs found in the lymph are not always accompanied by inflammation, which suggests that pEVs utilize lymph for mobilization and could be utilized as a non-inflammatory therapy when targeting cells within the lymphatics system [138,139]. pEVs’ ability to travel through the lymphatics in the absence of inflammation is a promising quality that needs to be investigated further.

Aside from the ability of pEVs to traffic across barriers, pEV therapy also has the potential to be applied to multiple diseases. This article focuses on viral diseases, but pEVs are present in multiple other diseases, such as cancer, rheumatoid arthritis, ischemic heart disease, and chronic wounds. In fact, some of these diseases have already experimentally employed EV therapy through platelet-rich plasma sourced EVs.

### 3.2. Platelet Rich Plasma as a Source for pEV Therapy

pEVs are typically isolated from plasma using differential centrifugation. One recent study used ultrafiltration and size exclusion chromatography to more specifically isolate EVs [140]. Many current methods use platelet-rich plasma (PRP) for the isolation of EVs generally (PRP-EVs), or specifically, exosomes (PRP-Exos) (Table 1) [141,142]. PRP is isolated from the plasma of individuals by centrifugation techniques and can be autologously reinfused into patients. PRP itself is a therapy used in the treatment of chronic wounds, among other diseases [143]. PRP treatment is thought to be relatively effective due to the growth factors released from activated platelets, such as platelet-derived growth factor (PDGF), transforming growth factor-b (TGF-b), and vascular endothelial growth factor (VEGF) [144]. PDGF, TGF-b, and VEGF play critical roles in tissue regeneration and wound healing; thus, it makes sense why PRP therapy has been found to be a successful treatment for chronic wounds [145,146,147]. Recently, PRP therapy has also been explored for the treatment of viral diseases, such as BK virus induced hemorrhagic cystitis, severe COVID-19, and for veterinary use in the treatment of ulcers caused by feline herpes virus-1 or canine herpes virus [148,149,150]. Another experimental platelet-based therapy is human platelet lysate (HPL), which involves the dissociation of platelet membranes to reveal various platelet factors, which has been used as an experimental therapy for COVID-19 [151,152,153,154]. This illustrates that the vesicle contents themselves are of therapeutic interest in treating viral diseases. Additionally, the success of HPL as an experimental treatment for COVID-19 pneumonia suggests that pEV therapy might be similarly successful in reducing viral disease complications.

PRP as a therapy by itself, is typically autologous, but it has been safely used allogeneically, as an adjuvant [155]. However, PRP derived EVs do not need to be autologous and could be generated in a different species [156]. Like exosomes, MVs are also a part of the EVs isolated using PRP. However, PRP-EV isolation would not be sufficient to distinguish between microvesicles, exosomes, and the other EV component apoptotic bodies [157]. Likely due to the lack of validated MV isolation protocols, there are limited studies focused specifically on PMVs in viral diseases. A recent in vitro HIV-1 study found that PMVs containing anti-HIV drugs are a safe method of anti-viral drug delivery [158]. Another in vitro study found that PRP-MVs increased wound healing [159]. A cancer study found that PMVs derived from platelet rich plasma promotes proliferation, migration, and osteogenic differentiation of bone mesenchymal stem cells in vitro, which suggests that PMVs play an important role in tissue regeneration [160].

PRP-EVs and PRP-Exos have been used in the treatment of several diseases and conditions, including cancer, pneumonia, atherosclerosis, rheumatoid arthritis, chronic wounds, muscle injury, and hemorrhagic shock (Table 1) [161,162,163,164,165,166,167]. Additionally, there are currently three clinicals trials utilizing PRP-Exos for chronic conditions, such as postoperative temporal bone cavity inflammation (CPTBCI) (NCT04281901), middle ear infection (NCT04761562), and lower back pain (NCT04849429) (Table 1). The CPTBCI trial is now complete, and found a reduction in patient symptoms, with 49% of patients remaining symptom-free for 12.7 months post-treatment (Table 1) [168].

Research involving methods to isolate pEVs from human plasma is growing, but there is still much to be determined, such as the importance of platelet activation stimuli [169,170]. pEVs are not yet widely studied in the context of virus diseases and many of the pEV experimental studies focus on diseases associated with chronic inflammation or disruptions in thrombosis/hemostasis. However, one study found that pEV therapy was successful in reducing cytokine storms during pneumonia in a mouse model, which has applications for severe IAV and SARS-CoV-2 complications (Table 1) [161]. Thus, whilst current treatments do not specifically target viral diseases, they can still be applicable to virus-specific pEV therapies.

Existing pEV therapies target conditions with manifestations that overlap with severe viral disease, which illustrates the potential for pEVs to be utilized in viral diseases. Additionally, pEVs and other released factors play crucial roles in disease progression as detailed above. Given the success of many recent pEV therapies and the potential application in viral diseases, we expect to see an increase in PRP derived pEV therapies targeting viral diseases in the near future.

### 3.3. In Vitro Engineering of EVs

Aside from PRP-generated pEVs, the feasibility of in vitro pEV generation is of great interest. In vitro generation of pEVs from human induced pluripotent stem cells (hiPSC) has been used for treatment of HER2+ breast-to-brain metastasis in a mouse model, although methods to generate the hiPSC-platelets can take up to 20 weeks (Table 1) [171]. Thus, while in vitro pEV generation is worth pursuing, current in vitro generation methods are time consuming. One study used in vivo generation of engineered pEVs by transplant of monoclonal antibody expressing platelets in a mouse model and in vivo activation (Table 1) [172]. Therefore, the method of generating pEVs from PRP is not necessarily the only method of generating therapeutic pEV and in vitro engineered platelets can generate pEVs in vivo. Through in vitro generation, EVs can be engineered to contain specific cargo or to express specific receptors (Figure 5). These techniques are called cargo-loading and surface engineering, respectively, and are already being used for non-platelet derived EVs. However, these techniques are applicable to pEVs given their similarities to other cell-derived EVs.

#### 3.3.1. Cargo-Loading of EVs

There are two main methods of loading cargo into EVs (exogenous and endogenous) and there are two main classes of molecules that can be loaded (small molecules/drugs and RNAs).

The two cargo loading techniques are exogenous loading and endogenous loading. In exogenous loading, EVs are collected and are either co-incubated, chemically transfected, electroporated, or sonicated in order to deliver the cargo into the EVs [173,174]. Hydrophobic molecules typically incorporate into EVs following co-incubation; however, hydrophilic molecules may need to utilize other techniques. Loading of hydrophilic agents can be done through pH-based chemical changes within the buffer to produce a pH gradient [173]. RNA-based cargo, such as small interfering RNA (siRNA), can be loaded into EVs using sonication or by physical approaches, such as electroporation [175]. In endogenous loading, the molecule or drug is incubated with the parent cells or parent cells are genetically engineered to overexpress specific RNA or proteins. Engineering targets for inducing miRNA and mRNA loading into EVs are hnRNPA2B1, Y-box protein 1, SYNCRIP, and ELVA protein HuR, which have various roles in RNA loading into EVs [176,177].

One therapeutic application of pEVs is for the encapsulation of small drugs/molecules. Current small drugs used are typically anti-inflammatory drugs, such as curcumin, piceatannol, aspirin, and dexamethasone [173,178,179]. These anti-inflammatory drugs can be accompanied by other therapeutic molecules, such as MHC-loaded peptides for T cell targeted expansion. Other than anti-inflammatory drugs, drugs that specifically reduce thrombosis in viral infections are also of interest such as therapeutic heparin (UFH or LMWH), which has been used for H1N1-induced pneumonia [59].

In addition to drugs, small molecules, such as MHCs loaded with peptides, show promising effects on inducing antigen specific T-cells. Dendritic EVs (OFexo) containing factors such as a curcumin analog and ovalbumin loaded onto MHC-II, have been shown to induce antigen specific regulatory T cells (Tregs) in mice [180]. This model for dendritic EVs is perhaps applicable to PMVs, which are known to carry MHC-I. It is possible that a CD8+ T cell targeted response could be induced by PMVs loaded with MHC-I and peptides. Alternatively, if PMVs can express MHC-II, which has not yet been shown, PMVs could be used to target the expansion of specific Tregs similar to the dendritic EVs.

RNAs, such as miRNA, siRNA and mRNA, can also be loaded into EVs and deliver specific genetic-based therapy to surrounding cells. RNA-based therapeutics are often thought of as a specific individualized therapy. Platelet microvesicles have been found to play a role in breast cancer disease malignancy processes [181]. In fact, platelet derived procoagulant microvesicles are shown to have a significant positive relationship with breast cancer metastasis and have been used to predict breast cancer metastasis in recent studies [182,183]. Finally, mRNA within EVs of implanted cells, reduced neurotoxicity in a model of Parkinson’s disease [184]. While there are limited studies on RNA encapsulation within EVs, these potential therapies are avenues of rapidly evolving research, which may be applicable to multiple diseases.

#### 3.3.2. Surface Engineering of EVs

Although the contents of EVs are incredibly important, it is equally as important to determine the surface markers on EVs. These surface markers can be used for migration, targeting specific tissues and cells, or for interactions with immune cells (Figure 5).

Surface engineering can be accomplished with cell engineering or chemical modification. Cell engineering of surface markers can be achieved by fusing a targeting sequence to EV-enriched membrane proteins or by inserting a target epitope into a protein domain [174,185,186]. Cell engineering is used to improve the targeting capacity, circulation time, and uptake of EVs by other cells. Chemical modification can include the use of click chemistry for the bioconjugation of molecules via covalent bonds and hybridization of EVs with synthetic liposomes [187,188]. Click chemistry is perhaps more stable, but the effects of click chemistry on the function of the protein must be evaluated. Hybridization of EVs with synthetic liposomes increases EV stability and half-life within circulation, whilst lowering immunogenicity [188].

The natural receptors present on EVs can be used for targeting certain cells. Natural receptors can be used to target specific cell uptake. Additionally, natural surface markers exhibit specific homing towards tissues such as the kidney and brain in the case of macrophage-derived EVs (Figure 5) [179,189]. Future studies might also utilize platelet-specific surface ligands on PMVs to increase targeting of immune cells or biologically isolated tissues. Further, natural enzymes such as phospholipase A_2_ are useful for inducing the release of cargo to the site of interest [190].

While natural surface expression is a simpler approach, engineering surface markers allows for more active targeting at known surface concentrations of the marker. Some of the markers that can be engineered are antibodies, peptide, small molecules, and aptamers (Figure 5). A novel synthetic multivalent antibody retargeted exosome (MART-Exo) expresses monoclonal antibodies specific for CD3 on T cells and epidermal growth factor receptor (EGFR) on cancer cells [191]. This antibody induces the cross-linking of T cells and EGFR-expressing breast cancer cells [191]. In addition to cell-specific targeting, EVs are engineered to have surface markers, such as CSTSMLKAC (IMTP), cardiac-targeting peptide (CTP), that can target the heart and cRGD peptides to target an ischemic brain (Figure 2) [192,193,194]. Inflammatory-targeting peptides are also of therapeutic interest but have not yet been confirmed to be effective. Anti-inflammatory therapy could also be employed by displaying multiple inflammatory cytokine receptors on the surface of the EV, to use the EVs as biological sponges for inflammatory cytokines (Figure 5).

### 3.4. Targeting Viruses with pEV Engineering

Given what we know about platelets and EV engineering, there are multiple potential therapies specific to viruses. Listed below are some specific applications of both cargo-loading and surface engineering for pEVs targeted at viruses.

#### 3.4.1. Anti-Viral Mediators in the Cargo of pEVs

MHC and peptides. The loading of MHC molecules with peptides could target specific T cells, inducing their effector functions against viruses.RNAs. EV RNAs have already been shown to exert effects on surrounding cells, influencing their production of molecules. Thus, using immunologically specific RNAs to alter immune cell phenotypes, is of great interest for viral disease.Anti-inflammatory mediators. Platelet-derived factors, such as sCD40L and components of α-granules (CXCL4), are of therapeutic interest due to their anti-inflammatory effects, specific targeting of immune cells, and role in viral disease progression.

#### 3.4.2. Anti-Viral pEV Surface Markers

Viral binding proteins. Engineered surface markers for viral binding (TLRs, CLEC-2, DC-SIGN, and ACE2) on the surface of pEVs is of potential therapeutic interest in order to prevent excessive platelet activation and thrombocytopenia. Viruses could be targeted to bind pEVs instead of platelets, just by the availability of engineered pEV surface molecules compared to platelets. In conjunction, these pEVs could be equipped with anti-viral molecules that release upon viral binding, preventing viral replication. In many severe viral diseases, thrombocytopenia is a biomarker of mortality; thus, by reducing platelet infection, perhaps platelets may more easily carry out their anti-viral effector functions.Sequestering inflammatory platelet mediators. Another application for engineering surface markers is to specifically target platelet-released factors such as sCD40L or inflammatory markers using monoclonal antibodies. This application would reduce an excess of inflammatory effects, which is often characteristic of severe disease. Severe viral infections often have elevated sCD40L and inflammatory cytokines. By sequestering these factors through antibody binding on pEVs, inflammatory positive feedback loops could be reduced.Homing to damaged tissues. PMV-like nanovesicles were engineered with the surface receptors CPIIb/IIIa and P-selectin in order to target the site of clot formation [190]. In viral diseases such as Dengue where the coagulation cascade can be disrupted, it might be beneficial to utilize receptors that home cells to the site of capillary leakage.

Together, both cargo loading and surface engineering could be effectively employed to specifically target viral diseases. However, before generating viral specific pEVs, it is extremely important to consider possible barriers to pEV generation and clinical application.

### 3.5. Considerations for pEV Therapy

pEV therapy has many promising applications, and various engineering techniques used by other EV therapies could be applied to pEV specific therapeutics. However, pEV therapy is quite novel and, without a doubt, there will be troubleshooting and adjustments to be made before pEV therapy is ready for clinical use. Some of the major considerations for the use of pEV therapy include biological barrier crossing, hypersensitivity, and the difficulties of working with sensitive cell fragments.

Although crossing biological barriers is a promising therapeutic attribute, it is also of concern if the crossing of biological barriers induces inflammation in areas where it could induce damage. In severe viral infections there is often inflammation and damage to organs or tissues that are normally isolated from severe inflammation, such as the brain, liver, and heart. In severe influenza cases, patients can exhibit damage within their heart, skeletal muscle, brain, or liver, but these areas do not contain the infectious virus [35,195,196,197,198,199]. In fact, the damage to these areas can in part be attributed to microvesicle release. The origin of the MVs is not clearly documented, but it is likely that some PMVs traffic to these areas and contribute to inflammation and damage. Future PMV therapeutics should investigate the various surface receptors to ensure the targeting of specific tissues and reduce off-target inflammatory effects.

Additionally, it is important to consider the possibility of hypersensitivity in nanoparticle-based therapies. Other nanoparticle therapies have been found to cause hypersensitivity reactions especially if administered intravenously [200]. These hypersensitivity reactions tend to activate the complement system and are determined to be complement-activation-related pseudoallergy (CARPA) [201].

Finally, another important consideration is that platelets and pEVs are difficult cell fragments to work with. Platelets are notoriously easily activated and difficult to cryopreserve, although a recent study was able to generate therapeutic pEVs from cryopreserved platelets [202]. In addition, platelets cannot replicate and are derived from megakaryocytes, which creates issues for in vitro generation of pEVs. Additionally, PMVs are nanoparticle in size and tricky to visualize within flow cytometry [203]. Therefore, pEV therapy might not be as easily employed as other cell-based therapies within cells that easily multiply in culture. But the idea of using platelet-based factors, receptors, and RNAs can still be applicable to other cell-based EVs while PMV-based therapy is still in its infancy.

Although there are difficulties associated with pEV therapy, the existence of current experimental pEVs and their success suggests that pEV therapy will emerge as a novel technique to treat many types of diseases, including viral diseases. As we learn more about platelet-released factors and their roles in diseases, therapeutics involving pEVs can be more easily generated and employed.

## 4. Conclusions

While there is still much to learn about platelet released factors, considerable progress is being made. Platelet released CD40L plays a role in viral disease progression, but sCD40L plasma levels are not always an indicator of severe disease for COVID-19. PMVs have diverse roles such as promoting inflammation, interacting with immune cells, and reprogramming cells with their encapsulated RNAs. α-granule contents also have varying and sometimes contradictory functions in hemostasis, vasculature, inflammatory effects, and anti-inflammatory effects. The diverse factors that platelets are capable of secreting, indicates the importance of understanding their functions to better understand exactly how platelets are involved in viral infections. Employing pEVs as a therapeutic delivery system for viral infections is still in its infancy, but we hope that by learning more about platelet-released factors we can determine specific therapeutic targets. By expanding our knowledge of platelet released mediators, we can discover new therapeutic molecules that could be encapsulated in pEVs to target hard to reach biological areas in viral diseases.

## Figures and Tables

**Figure 1 ijms-23-02321-f001:**
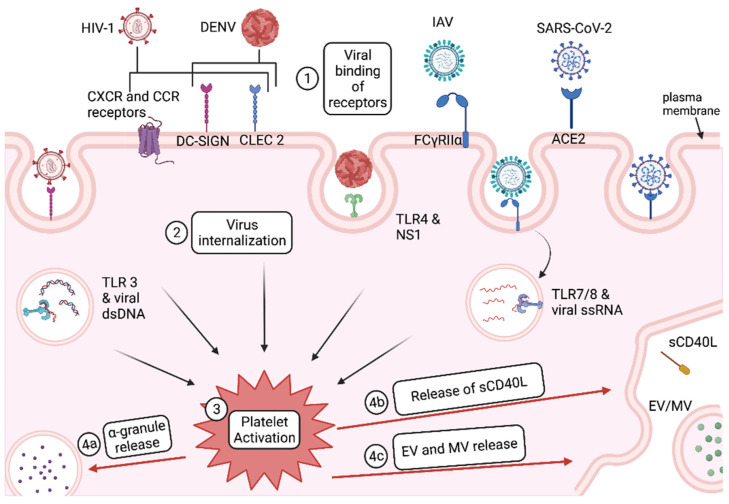
Viral entry and activation of platelets. 1. Human immunodeficiency virus-1 (HIV-1), dengue virus (DENV), influenza A virus (IAV), and severe acute respiratory syndrome coronavirus 2 (SARS-CoV-2) bind receptors on the platelet plasma membrane as well as within endosomes. Receptors involved include CXCR/CCR receptors, DC-SIGN, CLEC 2, FCγRIIα, ACE2, and TLRs 3/4/7/8. 2. Upon viral binding, viruses are internalized into the platelets. 3. Both viral binding and virus internalization lead to platelet activation. 4. Platelet activation results in platelet effector functions, such as 4a. α-granule release, 4b. release of soluble CD40L (sCD40L), and 4c. extracellular vesicle (EV) and microvesicle (MV) release. Created with BioRender.com (30 December 2021).

**Figure 2 ijms-23-02321-f002:**
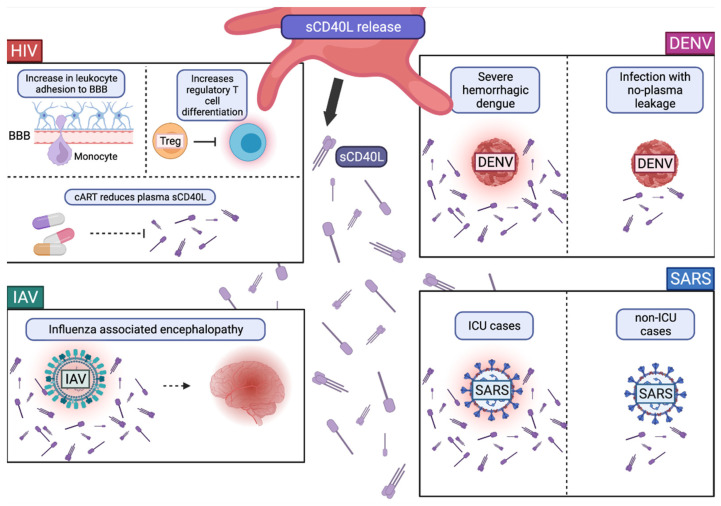
Role of sCD40L in Viral Infections. One major source of sCD40L in human plasma is platelets. In human immunodeficiency virus (HIV) infection, sCD40L increases leukocyte adhesion to the blood brain barrier (BBB), perhaps leading to extravasation and increases in regulatory T cell differentiation, which potentially leads to a reduction in T-cell responses. The administration of combination anti-retroviral therapy (cART) reduces plasma sCD40L levels. In dengue virus (DENV) infection, high levels of plasma sCD40L is associated with severe hemorrhagic dengue with plasma leakage. In influenza A virus (IAV), a high plasma sCD40L level is associated with severe disease and influenza-associated encephalopathy. In SARS-CoV-2 infection, high levels of plasma sCD40L was found in intensive care unit (ICU) cases compared to lower plasma sCD40L levels found in non-ICU cases. Created with BioRender.com (9 February 2022).

**Figure 3 ijms-23-02321-f003:**
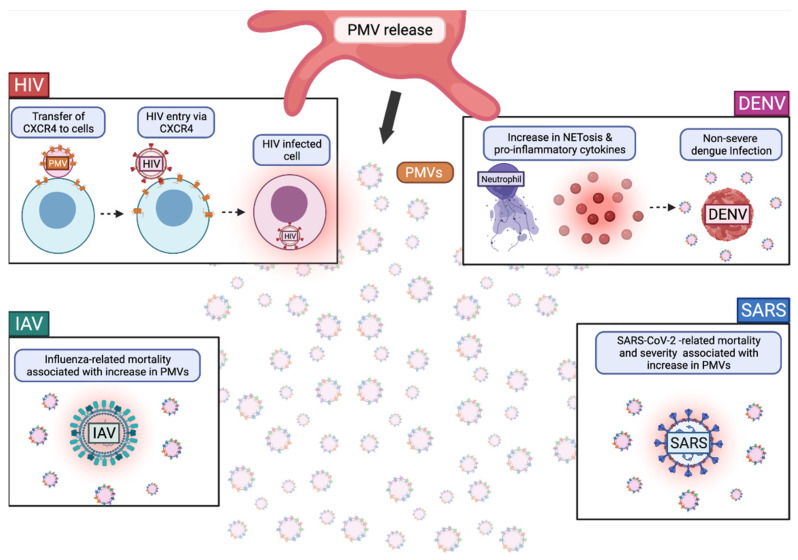
The Role of Platelet Microvesicles (PMVs) in Viral Infection. PMVs are a type of extracellular vesicle derived from activated platelets. In human immunodeficiency virus (HIV) infection, PMVs expressing CXCR4 have been shown in vitro to deliver CXCR4 upon fusion to other cells. Expressing CXCR4 on their surface, these cells are potentially more susceptible to HIV infection, perpetuating disease progression. In dengue virus (DENV) infection, PMVs are associated with increases in neutrophil functions, such as netosis as well as an increase in pro-inflammatory cytokines. Additionally, circulating PMVs are associated with non-severe DENV infection and are reduced in patients with severe disease, suggesting a beneficial role of PMVs in disease resolution. In both influenza A virus (IAV) and SARS-CoV-2 infections, mortality is associated with an increase in PMVs, suggesting a role of PMVs in promoting disease severity. Created with BioRender.com (9 February 2022).

**Figure 4 ijms-23-02321-f004:**
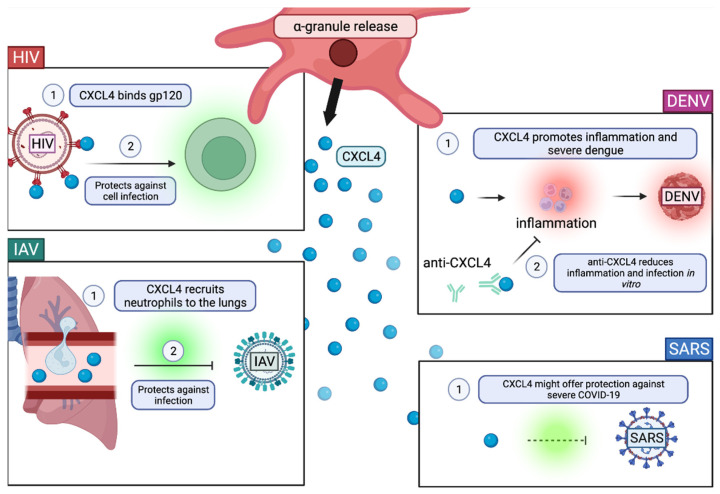
Role of α-Granules in Viral Infection. α-granule released from activated platelets contain CXCL4, a cytokine. In human immunodeficiency virus (HIV) infection, CXCL4 binds the major HIV surface protein gp120, which protects against cell infection. In dengue virus (DENV) infection, CXCL4 is known to promote inflammation and severe dengue. Anti-CXCL4 reduces inflammation and DENV infection in vitro. In influenza A virus (IAV) infection, CXCL4 recruits neutrophils to the lungs, which helps fight against infection. In SARS-CoV-2 infection, decreased plasma CXCL4 is correlated with poor disease progression, indicating that CXCL4 plays a protective role. Created with BioRender.com (9 February 2022).

**Figure 5 ijms-23-02321-f005:**
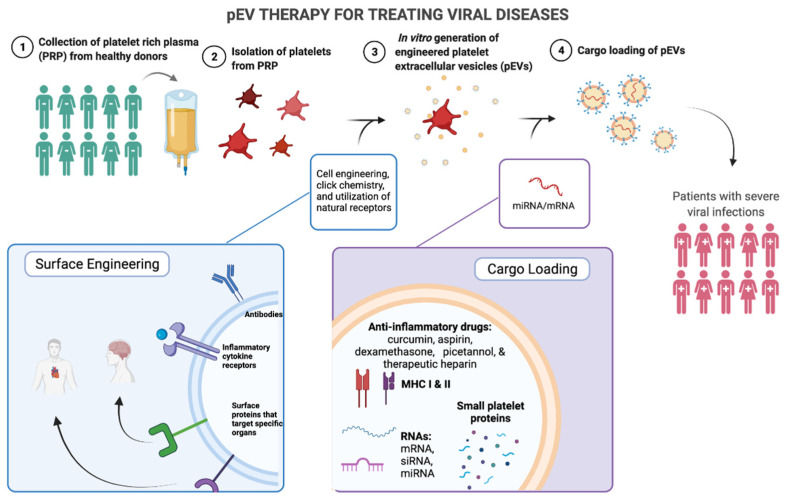
Platelet Extracellular Vesicle (pEV) Therapy for Treating Viral Diseases. 1. Platelet rich plasma (PRP) is isolated from healthy donors. 2. Platelets are isolated from PRP. 3. Surface engineering of platelets could involve the incorporation of receptors aimed at carrying out specific functions to treat a disease. In vitro culture methods are used to generate pEVs. 4. pEV cargo loading involves small drugs, molecules, and RNA. EV surface targeting includes antibodies, inflammatory cytokine receptors, and proteins that specifically target certain organs. After cargo loading, pEVs would be isolated to a clinical grade-level and administered to treat patients with severe viral infections. Created with BioRender.com (9 February 2022).

**Table 1 ijms-23-02321-t001:** Recent experimental pEV therapies for various diseases. Platelet extracellular vesicles (pEVs), platelet microparticles (PMPs), platelet microvesicles (PMVs), human induced pluripotent stem cells (hiPSC), platelet rich plasma (PRP), PRP derived exosomes (PRP-Exos), and platelet and extracellular vesicle rich plasma (PVRP).

Disease/ Condition	Model	Source of pEVs	Application to Viral Diseases	Reference (s)
HIV-1	In vitro Hela and U266 cells	Platelet concentrate, differential centrifugation. “PMPs”/PMVs	HIV and encapsulation of other anti-viral drugs for other viruses.	Soleymani, S. et al., 2019
Wound healing	In vitro HaCaT cell monolayers.	PRP-MVs	Viral diseases associated with aberrant hemostasis and thrombosis issues, such as dengue virus, Ebola, and severe SARS-CoV-2.	Lovisolo, F. et al., 2020
Cancer (HER2+ breast-to brain-metastasis)	Mouse	Indirect in vivo generation of pEVs via hiPSC-generated platelets infused into mice.	Cancer-inducing viral diseases.	Bhan, A. et al., 2021
Cancer	Mouse	Transplant of engineered platelets with monoclonal antibodies on their surface. In vivo generation of pEVs with these monoclonal antibodies upon platelet activation.	Cancer-inducing viral diseases.	Han, X. et al., 2019
Pneumonia	Mouse	Mouse PRP, PRP-EVs	Pulmonary inflammation diseases, such as influenza and SARS-CoV-2.	Ma, Q.; Fan, Q. et al., 2020
Atherosclerosis	Mouse	Mouse PRP, PRP-EVs		Ma, Q.; Fan, Q. et al., 2021
Rheumatoid Arthritis	Mouse	Mouse PRP, PRP-EVs		Ma, Q.; Bai, J. et al., 2021
Diabetes/chronic wounds	Rat	Human PRP, PRP-Exos	Viral diseases with aberrant hemostasis and thrombosis issues, such as dengue virus, Ebola, and severe SARS-CoV-2.	Guo, S. et al., 2017 Tao, S. et al., 2017
Muscle Injury	Rat	PRP-Exos		Iyer, S. et al., 2020
Hemorrhagic Shock	Rat (liver trauma)	PRP-EVs	Viral diseases with aberrant hemostasis and thrombosis issues, such as dengue virus, Ebola, and severe SARS-CoV-2.	Lopez, E. et al., 2019
Chronic Postoperative Temporal Bone Cavity Inflammation	Human Clinical Trials	PVRP		Vozel, D. et al., 2021 NCT04281901 (clinicaltrials.gov)
Chronic Middle Ear Infections	Human Clinical Trials	PVRP		NCT04761562 (clinicaltrials.gov)
Chronic Lower Back Pain	Human Clinical Trials	PRP-Exos		NCT04849429 (clinicaltrials.gov)

## Data Availability

Not applicable.

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
