# Peer review of "Platelet-Released Factors: Their Role in Viral Disease and Applications for Extracellular Vesicle (EV) Therapy"

_ijms, 2022, doi:10.3390/ijms23042321_

Round 1
Reviewer 1 Report
The paper of Ostermeier and colleagues summarize in a factual and reliable manner current knowledge on platelet immunobiology, effector functions and platelet interactions with specific viral diseases This paper may constitute a significant clinical value when consider platelet microvesicles as therapy for viral diseases especially for most recent severe acute respiratory syndrome coronavirus 2 (SARS-CoV-2). Since the presented results have been verified and published in the articles cited in the paper of Ostermeier et al., my role was to assess whether the authors managed to collect not only the most recent reports, but also to select the most appropriate and valuable ones. I my opinion authors have come to their conclusions based on the great papers in this topic. However, surprisingly, authors did not refer to the current literature but cited mostly old papers. It gives the impression that there are no new reports on this topic, which is hard to believe. Thus this review should be revised with more recent articles.
Regardless of the scientific significance of the publication's results I have raised some comments:
Comment 1: There are no full information in the affiliations (lack of full address, city and country names).
Comment 2: Figure 1 is not included in the paper although there is a legend to Fig. 1.
Comment 3: Figure 2 does not bring anything new to work especially considering that has not been created by authors.
Comment 4: Table 1 should be modified since it is to large and contains too detailed information in the present form.
Comment 5: There is a section/paragraph Discussion but in my opinion it is to short and authors did not discuss their own results in the contex of other reports. Since this is a review paper, Discussion should be replaced with section Summary or Conclusion.
Comment 6: Extensive editing of English language and style are required.
Author Response
RESPONSE: Thank you. In the revised version, we have now cited 116 articles, out of total 203 (58%), that were published in past 7 years. Of which, 74 articles (64%) were published in past 3 years. Our efforts are to cite the original articles, respectfully giving full credit to the pioneering work that was published decades ago, which has paved the way for current work cited in the manuscript. We believe, in this manner, we were able to highlight the advances (as well as roadblocks) in the field of viral diseases and the avenues for the applications of platelet-derived extracellular vesicle therapy thereof.
Regardless of the scientific significance of the publication's results I have raised some comments:
Comment 1: There are no full information in the affiliations (lack of full address, city and country names).
RESPONSE: We apologize for this error, which is now corrected.
Comment 2: Figure 1 is not included in the paper although there is a legend to Fig. 1.
RESPONSE: Thank you for pointing out this oversight, we have now included the Figure 1.
Comment 3: Figure 2 does not bring anything new to work especially considering that has not been created by authors.
RESPONSE: We have revised this Figure, and presented as Figure 5, to provide better overview of platelet-derived extracellular vesicle treatment modalities
Comment 4: Table 1 should be modified since it is too large and contains too detailed information in the present form.
RESPONSE: We agree. Table 1 is now simplified by removing column that summarizes “Outcomes”.
Comment 5: There is a section/paragraph Discussion but in my opinion it is too short and authors did not discuss their own results in the context of other reports. Since this is a review paper, Discussion should be replaced with section Summary or Conclusion.
RESPONSE: Thank you. The discussion is now replaced with Conclusions.
Comment 6: Extensive editing of English language and style are required.
RESPONSE: Our apologies. We have conducted extensive grammatical editing in the revised version of this manuscript.
Reviewer 2 Report
The Manuscript by Ostermeier et al; is a concise review, relevant to the field topic to review. The study describes the “Platelet-Released Factors: Their Role in Viral Disease and Applications for Extracellular Vesicle (EV) Therapy”. They reviewed Platelets, Platelet Activation, Virus Entry and Activation in Platelets. Then, Platelet Effector Functions. Finally, about Therapeutic Applications and Conclusion. Overall, the review is good and requires minor revisions before acceptance.
- Background requires info about EVs.
- Table 1 is informative.
- Adding Figures with illustrations are required to make the review impactful. Such as Platelet’s isolation, formation, EVs biogenesis, Therapy based on Platelets. etc
- Change 4. Discussion to 4. Conclusion
Author Response
Thank you.
- Background requires info about EVs.
RESPONSE: Thank you for suggesting, we have now included a brief introduction to extracellular vesicles in the ‘Background’ section.
- Table 1 is informative.
RESPONSE: Thank you. As outlined above, we have simplified Table 1 while retaining the information about the outcome of the recent experimental platelet-derived extracellular vesicle therapies in the text.
- Adding Figures with illustrations are required to make the review impactful. Such as Platelet’s isolation, formation, EVs biogenesis, Therapy based on Platelets. Etc.
RESPONSE: We agree with the Reviewer, and in response to this important suggestion we have included 3 new figures to illustrate the 3 different platelet-derived factors that are implicated in viral diseases.
- Change 4. Discussion to 4. Conclusion
RESPONSE: Thank you. The discussion is now replaced with Conclusions.
Round 2
Reviewer 1 Report
Authors have responded to all my comments and made corrections in the manuscript according to my suggestions. Thus I do not see objections against accepting it in the present form.